# Overcoming Catastrophic Interference using Conceptor-Aided Backpropagation

**Xu He, Herbert Jaeger**
Department of Computer Science and Electrical Engineering
Jacobs University Bremen
Bremen, 28759, Germany
`{x.he,h.jaeger}@jacobs-university.de`

## Abstract

Catastrophic interference has been a major roadblock in the research of continual learning. Here we propose a variant of the back-propagation algorithm, "conceptor-aided backprop" (CAB), in which gradients are shielded by conceptors against degradation of previously learned tasks. Conceptors have their origin in reservoir computing, where they have been previously shown to overcome catastrophic forgetting. CAB extends these results to deep feedforward networks. On the disjoint and permuted MNIST tasks, CAB outperforms two other methods for coping with catastrophic interference that have recently been proposed.

## 1 Introduction

Agents with general artificial intelligence are supposed to learn and perform well on multiple tasks. Continual learning refers to the scenarios where a machine learning system can retain previously acquired skills while learning new ones. However, when trained on a sequence of tasks, neural networks usually forget about previous tasks after their weights are adjusted for a new task. This notorious problem known as *catastrophic interference* (CI) (McCloskey & Cohen, 1989; Ratcliff, 1990; French, 1999; Kumaran et al., 2016) poses a serious challenge towards continual learning.

Many approaches have been proposed to overcome or mitigate the problem of CI in the last three decades (Hinton & Plaut, 1987; French, 1991; Ans & Rousset, 1997; French, 1997; Srivastava et al., 2014). Especially recently, an avalanche of new methods in the deep learning field has brought about dramatic improvements in continual learning in neural networks. Kirkpatrick et al. (2017) introduced a regularization-based method called *elastic weight consolidation* (EWC), which uses the posterior distribution of parameters for the old tasks as a prior for the new task. They approximated the posterior by a Gaussian distribution with the parameters for old tasks as the mean and the inverse diagonal of the Fisher information matrix as the variance. Lee et al. (2017) introduced two *incremental moment matching* (IMM) methods called mean-IMM and mode-IMM. Mean-IMM approximates the distribution of parameters for both old and new tasks by a Gaussian distribution, which is estimated by minimizing its KL-divergence from the mixture of two Gaussian posteriors, one for the old task and the other one for the new task. Mode-IMM estimates the mode of this mixture of two Gaussians and uses it as the optimal parameters for both tasks.

In the field of Reservoir Computing (Jaeger, 2001; Maass et al., 2002), an effective solution to CI using *conceptors* was proposed by Jaeger (2014) to incrementally train a recurrent neural network to generate spatial-temporal signals. Conceptors are a general-purpose neuro-computational mechanism that can be used in a diversity of neural information processing tasks including temporal pattern classification, one-shot learning, human motion pattern generation, de-noising and signal separation (Jaeger, 2017). In this paper, we adopt and extend the method introduced in Jaeger (2014) and propose a *conceptor-aided backpropagation* (CAB) algorithm to train feed-forward networks. For each layer of a network, CAB computes a conceptor to characterize the linear subspace spanned by the neural activations in that layer that have appeared in already learned tasks. When the network is trained on a new task, CAB uses the conceptor to adjust the gradients given by backpropagation so that the linear transformation restricted to the characterized subspace will be preserved after the

gradient descent procedure. Experiment results of two benchmark tests showed highly competitive performance of CAB.

The rest of this paper is structured as follows. Section 2 introduces conceptors and their application to incremental learning by ridge regression. Section 3 extends the method to stochastic gradient descent and describes the CAB algorithm. Section 4 compares its performance on the permuted and disjoint MNIST tasks to recent methods that address the same problem. Finally we conclude our paper in Section 5.

## 2 INCREMENTAL RIDGE REGRESSION BY CONCEPTORS

This section reviews the basics of conceptor theory and its application to incrementally training linear readouts of recurrent neural networks as used in reservoir computing. A comprehensive treatment can be found in (Jaeger, 2014).

### 2.1 CONCEPTORS

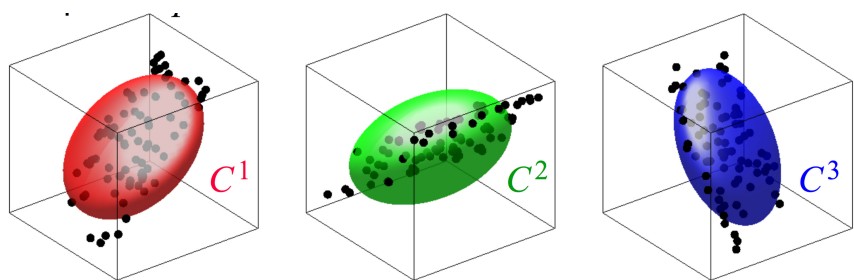

Figure 1: 3D point clouds (black dots) and their corresponding conceptors, represented by ellipsoids whose axes are the singular vectors of conceptors and the lengths of these axes match the singular values of conceptors. Each edge of the plot boxes range from $-1$ to $+1$ admitted by neural dynamics with a $tanh$ nonlinearity; conceptor ellipsiods lie inside the unit sphere.

In brief, a *matrix conceptor* $C$ for some vector-valued random variable $x \in \mathbb{R}^N$ is defined as a linear transformation that minimizes the following loss function.

$$\mathbb{E}_x[||x - Cx||^2] + \alpha^{-2}||C||^2_{\text{fro}} \tag{1}$$

where $\alpha$ is a control parameter called *aperture* and $|| \cdot ||_{\text{fro}}$ is the Frobenius norm. This optimization problem has a closed-form solution

$$C = R(R + \alpha^{-2}I)^{-1} \tag{2}$$

where $R = \mathbb{E}_x[xx^\top]$ is the $N \times N$ correlation matrix of $x$, and $I$ is the $N \times N$ identity matrix. This result given in (2) can be understood by studying the singular value decomposition (SVD) of $C$. If $R = U\Sigma U^\top$ is the SVD of $R$, then the SVD of $C$ is given as $USU^\top$, where the singular values $s_i$ of $C$ can be written in terms of the singular values $\sigma_i$ of $R$: $s_i = \sigma_i/(\sigma_i + \alpha^{-2}) \in [0, 1)$. In intuitive terms, $C$ is a soft projection matrix on the linear subspace where the samples of $x$ lie. For a vector $y$ in this subspace, $C$ acts like the identity: $Cy \approx y$, and when some noise $\epsilon$ orthogonal to the subspace is added to $y$, $C$ de-noises: $C(y + \epsilon) \approx y$. Figure 1 shows the ellipsoids corresponding to three sets of $\mathbb{R}^3$ points. We define the quota $Q(C)$ of a conceptor to be the mean singular values: $Q(C) := \frac{1}{N}\sum_{i=1}^N s_i$. Intuitively, the quota measures the fraction of the total dimensions of the entire vector space that is claimed by $C$.

Moreover, logic operations that satisfy most laws of Boolean logic can be defined on matrix conceptors as the following:

$$\neg C := I - C, \tag{3}$$

$$C^i \lor C^j := (R^i + R^j)(R^i + R^j + \alpha^{-2}I)^{-1} \tag{4}$$

$$C^i \land C^j := \neg(\neg C^i \lor \neg C^j) \tag{5}$$

where $\neg C$ softly projects onto a linear subspace that can be roughly understood as the orthogonal complement of the subspace characterized by $C$. $C^i \vee C^j$ is the conceptor computed from the union of the two sets of sample points from which $C^i$ and $C^j$ are computed. It describes a space that is approximately the sum of linear subspaces characterized by $C^i$ and $C^j$, respectively. The definition of $C^i \wedge C^j$ reflects de Morgan's law. Figure 2 illustrates the geometry of these operations.

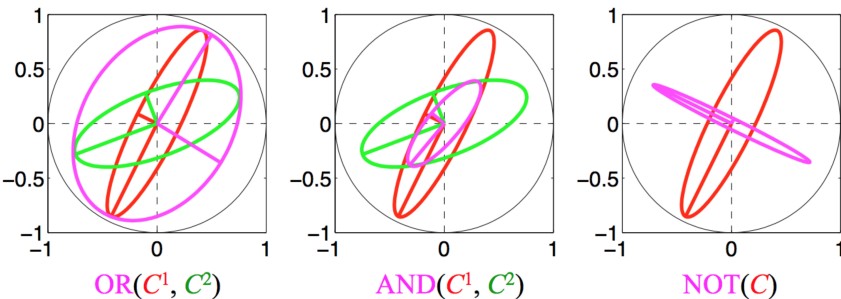

Figure 2: Geometry of Boolean operations on 2-dimensional conceptors. The OR (resp. AND) operation gives a conceptor whose ellipsoid approximately is the smallest (largest) ellipsoid enclosing (contained in) the argument conceptor's ellipsoids.

## 2.2 INCREMENTAL RIDGE REGRESSION

This subsection explains how conceptors can be applied to master continual learning in a simple linear model trained on a supervised task by ridge regression. The training is done sequentially on multiple input-to-output mapping tasks. This simplified scenario illustrates the working principle of continual learning with conceptors and will later be used repeatedly as a sub-procedure in the CAB algorithm for training multilayer feed-forward networks.

Consider a sequence of $m$ incoming tasks indexed by $j$. We denote the training dataset for the $j$-th task by $\{(x_1^j, y_1^j), \cdots, (x_n^j, y_n^j)\}$, where $x_i^j \in \mathbb{R}^N$ are input vectors and $y_i^j \in \mathbb{R}^M$ their corresponding target outputs. Whenever the training dataset for a new task is available, the incremental learning method will compute a matrix conceptor $C^j$ for the input variable of the new task using Equation 2 and update the linear model, resulting in a sequence of linear models $W^1, \ldots W^m$ such that $W^j$ solves not only the $j$-th task but also all previous tasks: for $k \leq j$, $y^k \approx W^j x^k$. The conceptor $C^j$ is a soft projection matrix onto the linear subspace spanned by input patterns from the $j$-th task. Then, $A^{j-1} = C^1 \vee \cdots \vee C^{j-1}$ characterizes the memory space already claimed by the tasks $1, \ldots, j-1$ and $F^j = \neg A^{j-1}$, the orthogonal complement of $A^j - 1$, represents the memory space still free for the $j$-th task. Here "memory space" refers to the linear space of input vectors. In detail, this method proceeds in the following way:

- **Initialization (no task trained yet):** $W^0 = 0_{M \times N}, A^0 = 0_{N \times N}$.
- **Incremental task learning:** For tasks $j = 1, \ldots, m$ do:
  1. Store the input vectors from the $j$-th training dataset of size $n$ into a $N \times n$ sized input collection matrix $X^j$, and store the output vectors into a $M \times n$ sized output collection matrix $Y^j$.
  2. Compute the conceptor for this task by $C^j = R^j(R^j + \alpha^{-2}I)^{-1}$, where $R^j = \frac{1}{n} X^j X^{j\top}$
  3. Train an increment matrix $W_{inc}^j$ (to be added to $W^{j-1}$, yielding $W^j$), with the crucial aid of a helper conceptor $F^j$:
     (a) $F^j := \neg A^{j-1}$ (*comment: this conceptor characterizes the "still disposable" memory space for the $j$-th task*),
     (b) $T := Y^j - (W^{j-1}X^j)$ (*comment: this matrix consists of target values for a linear regression to compute $W_{inc}^j$*),
     (c) $S := F^j X^j$ (*comment: this matrix consists of input arguments for the linear regression*),

(d) $W_{inc}^j = ((SS^\top/n + \lambda^{-2}I)^{-1}ST^\top/n)^\top$ (*comment: carry out the regression, regularized by $\lambda^{-2}$*),

4. Update $W^j$: $W^j = W^{j-1} + W_{inc}^j$.

5. Update $A : A^j = A^{j-1} \vee C^j$ (*comment: this is possible due to the associativity of the $\vee$ operation on conceptors*)

The weight increment $W_{inc}^j$ does not interfere much with the previously learned weights $W^{j-1}$ because the regularization in step 3(d) constrains the row space of $W_{inc}^j$ to be only the linear subspace spanned by input arguments defined in 3(c), which are inside the kernel of $W^{j-1}$ due to the projection by $F^j$. Intuitively speaking, when learning a new task, this algorithm exploits only the components of input vectors in the still unused space (kernel of $W^{j-1}$, characterized by $F^j$) to compensate errors for the new task and leaves the directions in the already used memory space (row space of $W^{j-1}$, characterized by $A^{j-1}$) intact.

# 3 CONCEPTOR-AIDED SGD AND BACK-PROP

In this section, we first derive a stochastic gradient descent version of the algorithm described in the previous section, then present the procedure of CAB.

## 3.1 SGD

In the algorithm introduced in the previous section, $W_{inc}^j$ is computed by ridge regression, which offers a closed-form solution to minimize the following cost function

$$\mathcal{J}(W_{inc}^j) := \mathbb{E}[|W_{inc}^j s - t|^2] + \lambda^{-2}|W_{inc}^j|_{\text{fro}}^2 \tag{6}$$

where $t = y^j - W^{j-1}x^j, s = F^j x^j$. One can also minimize this cost function by stochastic gradient descent (SGD), which starts from an initial guess of $W_{inc}^j$ and repeatedly performs the following update

$$W_{inc}^j \leftarrow W_{inc}^j - \eta \nabla_{W_{inc}^j} \mathcal{J}(W_{inc}^j) \tag{7}$$

where $\eta$ is the learning rate and the gradient is given by:

$$\nabla_{W_{inc}^j} \mathcal{J}(W_{inc}^j) = 2\mathbb{E}[(W_{inc}^j s - t)s^\top] + 2\lambda^{-2}W_{inc}^j \tag{8}$$

Substituting $t$ by $y^j - W^{j-1}x^j$ and $s$ by $F^j x^j = (I - A^{j-1})x^j$ in (8), we get

$$\nabla_{W_{inc}^j} \mathcal{J}(W_{inc}^j) = 2\mathbb{E}[(W_{inc}^j(I - A^{j-1})x^j - y^j + W^{j-1}x^j)s^\top] + 2\lambda^{-2}W_{inc}^j \tag{9}$$

$$= 2\mathbb{E}[(-W_{inc}^j A^{j-1}x^j + (W^{j-1} + W_{inc}^j)x^j - y^j)s^\top] + 2\lambda^{-2}W_{inc}^j \tag{10}$$

Due to the regularization term in the cost function, as the optimization goes on, eventually $W_{inc}$ will null the input components that are not inside the linear subspace characterized by $F^j$, hence $W_{inc}^j A^{j-1}x^j$ will converge to 0 as the algorithm proceeds. In addition, since $W^j = W^{j-1} + W_{inc}^j$, (10) can be simplified to

$$\nabla_{W_{inc}^j} \mathcal{J}(W_{inc}^j) = 2\mathbb{E}[(W^j x^j - y^j)s^\top] + 2\lambda^{-2}W_{inc}^j \tag{11}$$

Adding $W^{j-1}$ to both sides of (7), we obtain the update rule for $W^j$:

$$W^j \leftarrow W^j - 2\eta\mathbb{E}[es^\top] + 2\eta\lambda^{-2}W_{inc}^j \tag{12}$$

where $e := W^j x^j - y^j$. In practice, at every iteration, the expected value can be approximated by a mini-batch of size $n_B$, indexed by $i_B$:

$$\hat{\mathbb{E}}[es^\top] = \frac{1}{n_B}\sum_{i_B=0}^{L}(W^j x_{i_B}^j - y_{i_B}^j)(F^j x_{i_B}^j)^\top = \frac{1}{n_B}\sum_{i_B=0}^{L}(W^j x_{i_B}^j - y_{i_B}^j)x_{i_B}^{j\top}F^j \tag{13}$$

where the transpose for $F^j$ can be dropped since it is symmetric.

If we only train the $j-$th task without considering the previous tasks, the update rule given by normal SGD is

$$W^j \leftarrow W^j - 2\eta\mathbb{E}[ex^{j\top}] + 2\eta\lambda^{-2}W^j \tag{14}$$

Comparing this to the update rule in (12), we notice two modifications when a conceptor is adopted to avoid CI: first, the gradient of weights are calculated using the conceptor-projected input vector $s = F^j x^j$ instead of the original input vector $x^j$; second, regularization is done on the weight increment $W_{inc}^j$ rather than the final weight $W^j$. These two modifications lead to our design of the conceptor-aided algorithm for training multilayer feed-forward networks.

## 3.2 BACKPROP

The basic idea of CAB is to guide the gradients of the loss function on every linear component of the network by a matrix conceptor computed from previous tasks during error back-propagation (Rumelhart et al., 1986), repeatedly applying the conceptor-aided SGD technique introduced in the previous section in every layer.

Consider a feed-forward network with $L + 1$ layers, indexed by $l = 0, \ldots L$, such that the 0-th and the $L$-th layers are the input and output layers respectively. $W^{(l)}$ represents the linear connections between the $(l-1)$-th and the $l$-th layer, where we refer to the former as the pre-synaptic layer with respect to $W^{(l)}$, and to the latter as the post-synaptic layer. We denote by $N^{(l)}$ the size of the $l$-th layer (excluding the bias unit) and $A^{(l)^j}$ a conceptor characterizing the memory space in the $l$-th layer used up by the first $j$ tasks. Let $\sigma(\cdot)$ be the activation function of the nonlinear neurons and $\theta$ all the parameters of the network to be trained. Then the incremental training method with CAB proceeds as follows:

- **Initialization (no task trained yet):** $\forall l = 0, \ldots, L - 1, A^{(l)^0} := 0_{(N^{(l)}+1) \times (N^{(l)}+1)}$, and randomly initialize $W^{(l+1)^0}$ to be a matrix of size $N^{(l+1)} \times (N^{(l)} + 1)$.

- **Incremental task learning:** For $j = 1, \ldots, m$ do:

  1. $\forall l = 0, \ldots, L - 1, F^{(l)^j} = \neg A^{(l)^{(j-1)}}$. (*This conceptor characterizes the still disposable vector space in layer $l$ for learning task $j$*)

  2. Update the network parameters $\theta^{(j-1)}$ obtained after training the first $j - 1$ tasks to $\theta^j$ by stochastic gradient descent, where the gradients are computed by CAB instead of the classical backprop. Algorithms 1 and 2 detail the forward and backward pass of CAB, respectively. Different from classical backprop, the gradients are guided by a matrix conceptor $F^{(l)^j}$, such that in each layer only the activity in the still disposable memory space will contribute to the gradient. Note that the conceptors remain the same until convergence of the network for task $j$.

  3. After training on the $j$-th task, run the forward procedure again on a batch of $n_B$ input vectors, indexed by $i_B$, taken from the $j$-th training dataset, to collect activations $h_{i_B}^{(l)^j}$ of each layer into a $N^{(l)} \times n_B$ sized matrix $H^{(l)^j}$, and set the correlation matrix $R^{(l)^j} = \frac{1}{n_B} H^{(l)^j} (H^{(l)^j})^\top$.

  4. Compute a conceptor on the $l$-th layer for the $j$-th pattern by $C^{(l)^j} = R^{(l)^j}(R^{(l)^j} + \alpha^{-2}I_{N^{(l)} \times N^{(l)}})^{-1}, \forall l = 0, \ldots, L - 1$. Finding an optimal aperture can be done by a cross-validation search[1].

  5. Update the conceptor for already used space in every layer: $A^{(l)^j} = A^{(l)^j} \vee C^{(l)^j}, \forall l = 0, \ldots, L - 1$.

---

**Algorithm 1** The forward procedure of conceptor-aided backprop, adapted from the traditional backprop. Input vectors are passed through a feed-forward network to compute the cost function. $\mathcal{L}(\hat{y}^j, y^j)$ denotes the loss for the $j$-th task, to which a regularizer $\Omega(\theta_{inc}^j) = \Omega(\theta^j - \theta^{j-1}) = ||\theta^j - \theta^{j-1}||_{\text{fro}}^2$ is added to obtain the total cost $\mathcal{J}$, where $\theta$ contains all the weights (biases are considered as weights connected to the bias units). The increment of parameters rather than the parameters themselves are regularized, similar to the conceptor-aided SGD.

---

**Require:** Network depth, $l$

**Require:** $W^{(l)j}, l \in \{1, \dots, L\}$, the weight matrices of the network

**Require:** $x^j$, one input vector of the $j$-th task

**Require:** $y^j$, the target output for $x^j$

1:  $h^{(0)} = x^j$
2:  **for** $l = 1, \dots L$ **do**
3:      $b^{(l)} = [h^{(l-1)\top}, 1]^\top$, include the bias unit
4:      $a^{(l)} = W^{(l)j} b^{(l)}$
5:      $h^{(l)} = \sigma(a^{(l)})$
6:  **end for**
7:  $\hat{y}^j = h^{(l)}$
8:  $\mathcal{J} = \mathcal{L}(\hat{y}^j, y^j) + \lambda \Omega(\theta_{inc}^j)$

---

**Algorithm 2** The backward procedure of conceptor-aided backprop for the $j$-th task, adapted from the traditional backprop. The gradient $g$ of the loss function $\mathcal{L}$ on the activations $a^{(l)}$ represents the error for the linear transformation $W^{(l)j}$ between the $(l-1)$-th and the $l$-th layers. In the standard backprop algorithm, the gradient of $\mathcal{L}$ on $W^{(l)j}$ is computed as an outer product of the post-synaptic errors $g$ and the pre-synaptic activities $h^{(l-1)}$. This resembles the computation of the gradient in the linear SGD algorithm, which motivates us to apply conceptors in a similar fashion as in the conceptor-aided SGD. Specifically, we project the gradient $\nabla_{W^{(l)j}} \mathcal{L}$ by the matrix conceptor $F^{(l-1)j}$ that indicates the free memory space on the pre-synaptic layer.

---

1:
$$g \leftarrow \nabla_{\hat{y}} \mathcal{J} = \nabla_{\hat{y}} \mathcal{L}(\hat{y}, y)$$

2:  **for** $l = L, L-1, \dots, 1$ **do**
3:      Convert the gradient on the layer's output into a gradient on the pre-nonlinearity activation ($\odot$ denotes element-wise multiplication):
$$g \leftarrow \nabla_{a^{(l)}} \mathcal{J} = g \odot \sigma'(a^{(l)})$$

4:      Compute the gradient of weights, project it by $F^{(l-1)j}$, and add it to the regularization term on the increment:
$$\begin{aligned} \nabla_{W^{(l)j}} \mathcal{J} &= g(F^{(l-1)j} b^{(l-1)})^\top + \lambda \nabla_{W^{(l)j}} \Omega(\theta_{inc}^j) = g b^{(l-1)\top} F^{(l-1)j} + 2\lambda W_{inc}^{(l)j} \\ &= g b^{(l-1)\top} F^{(l-1)j} + 2\lambda (W^{(l)j} - W^{(l)j-1}) \end{aligned}$$

5:      Propagate the gradients w.r.t. the next lower-level hidden layers activations:
$$g \leftarrow \nabla_{h^{(l-1)}} \mathcal{J} = {W^{(l)j}}^\top g$$

6:  **end for**

---

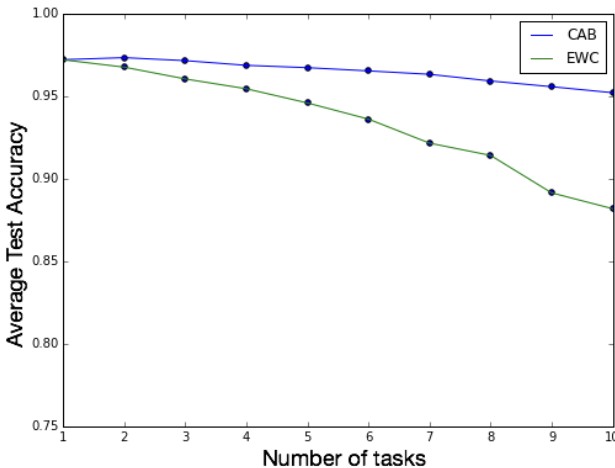

Figure 3: Average performance across already learned permuted MNIST tasks using CAB or EWC

## 4 EXPERIMENTS

### 4.1 PERMUTED MNIST EXPERIMENT

To test the performance of CAB, we evaluated it on the permuted MNIST experiment (Srivastava et al., 2013; Goodfellow et al., 2014; Kirkpatrick et al., 2017; Lee et al., 2017), where a sequence of pattern recognition tasks are created from the MNIST dataset (LeCun et al., 1998). For each task, a random permutation of input image pixels is generated and applied to all images in MNIST to obtain a new shuffled dataset, equally difficult to recognize as the original one, the objective of each task is to recognize these images with shuffled pixels.

For a proof-of-concept demonstration, we trained a simple but sufficient feed-forward network with [784-100-10] of neurons to classify 10 permuted MNIST datasets. The network has logistic sigmoid neurons in both hidden and output layers, and is trained with mean squared error as the cost function. Vanilla SGD was used in all experiments to optimize the cost function. Learning rate and aperture were set to 0.1 and 4, respectively. For comparison, we also tested EWC on the same task with the same network architecture, based on the implementation by Seff (2017). The parameters chosen for the EWC algorithm were 0.01 for the learning rate and 15 for the weight of the Fisher penalty term. Figure 3 shows the performance of CAB on this task, the average testing accuracy is 95.2% after learning all 10 tasks sequentially. Although a fair amount of effort was spent on searching for optimal parameters for EWC, the accuracies shown here might still not reflect its best performance. However, the same experiment with EWC was also conducted in Kemker et al. (2017), where the authors reimplemented EWC on a network with higher capacity (2 hidden layers and 400 ReLU neurons per layer) and the resulting average accuracy after learning 10 tasks sequentially was shown to be around 93%.

Since all tasks are generated by permuting the same dataset, the portion of the input space occupied by each of them should have the same size. However, as more tasks are learned, the chance that the space of a new task will overlap with the already used input space increases. Figure 4 shows the singular value spectra and quota of the input and hidden layer conceptors every time after a new task is learned. As the incremental learning proceeds, it becomes less likely for a new task to be in the free space. For example, the second task increases the quota of the input layer memory space by 0.1, whereas the 10th task increases it by only 0.03. However, CAB still manages to make the network learn new tasks based on their input components in the non-overlapping space.

---

[1] Jaeger (2014) proposes a number of methods for analytical aperture optimization. It remains for future work to determine how these methods transfer to our situation.

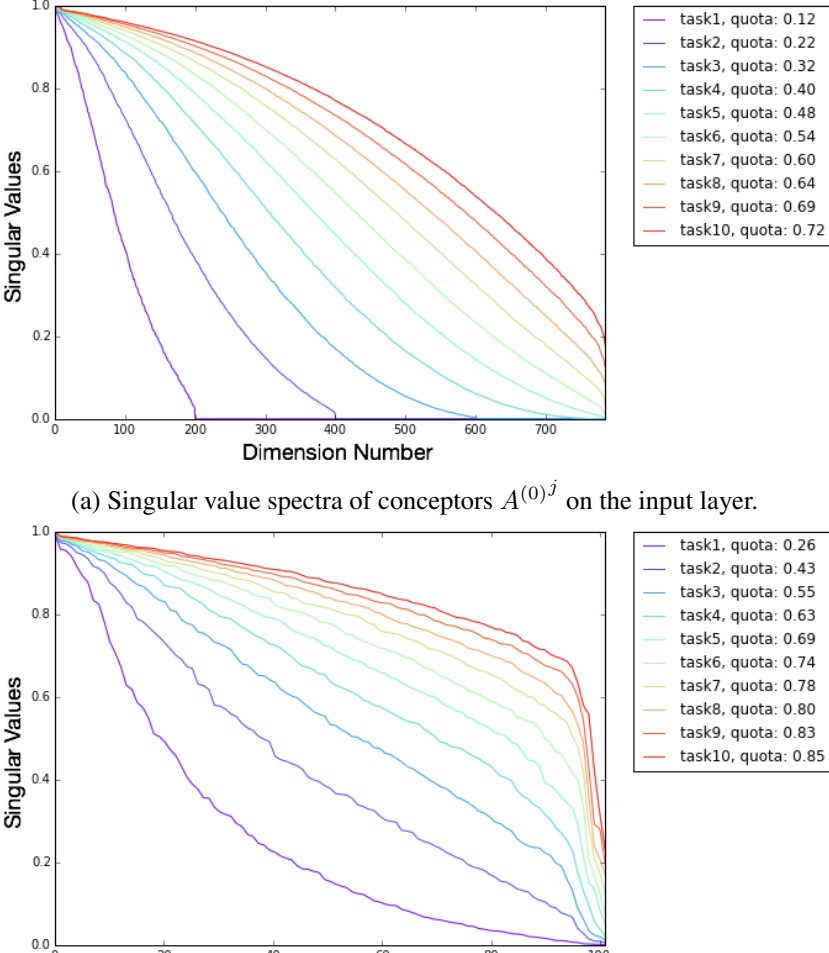

(a) Singular value spectra of conceptors $A^{(0)^j}$ on the input layer.

(b) Singular value spectra of conceptors $A^{(1)^j}$ on the hidden layer.

Figure 4: The development of singular value spectra of conceptors for "used-up" space on the input layer and hidden layer during incremental learning of 10 permuted MNIST tasks. Quota of these conceptors are displayed in the legends.

## 4.2 DISJOINT MNIST EXPERIMENT

We then applied CAB to categorize the disjoint MNIST datasets into 10 classes (Srivastava et al., 2013; Lee et al., 2017). In this experiment, the original MNIST dataset is divided into two disjoint datasets with the first one consisting of data for the first five digits (0 to 4), and the second one of the remaining five digits (5 to 9). This task requires a network to learn these two datasets one after the other, then examines its performance of classifying the entire MNIST testing images into 10 classes. The current state-of-the-art accuracy on this task, averaged over 10 learning trials, is $94.12(\pm0.27)\%$, achieved by Lee et al. (2017) using IMM. They also tested EWC on the same task and the average accuracy was $52.72(\pm1.36)\%$.

To test our method, we trained a feed-forward network with [784-800-10] neurons. Logistic sigmoid nonlinearities were used in both hidden and output layers, and the network was trained with vanilla SGD to minimize mean squared errors. The aperture $\alpha = 9$ was used for all conceptors on all layers, learning rate $\eta$ and regularization coefficient $\lambda$ were chosen to be $0.1$ and $0.005$ respectively. The accuracy of CAB on this task, measured by repeating the experiment 10 times, is $94.91(\pm0.30)\%$. It is worth mentioning that the network used by Lee et al. (2017) for testing IMM and EWC had [784-800-800-10] rectified linear units (ReLU), so CAB achieved better performance with fewer layers and neurons.

## 4.3 COMPUTATIONAL COST

If a conceptor is computed by ridge regression, the time complexity is $O(nN^2 + N^3)$ when the design matrix is dense, where $n$ is the number of samples and $N$ the number of features. In terms of wall time measures, the time taken to compute a conceptor from the entire MNIST training set (in this case, $n = 55000$ images and $N = 784$ pixels, corresponding to the input layer in our networks) is $0.42$ seconds of standard notebook CPU time on average. Although we did not implement it in these experiments, incremental online adaptation of conceptors by gradient descent is also possible in principle and would come at a cost of $O(N^2)$ per update.

## 5 CONCLUSION

In this work, we first reviewed the conceptor-based incremental ridge regression algorithm, introduced in section 3.11 of Jaeger (2014) for memory management in recurrent neural networks. Then we derived its stochastic gradient descent version for optimizing the same objective. Finally we designed a conceptor-aided backprop algorithm by applying a conceptor to every linear layer of a feed-forward network. This method uses conceptors to guide gradients of parameters during the backpropagation procedure. As a result, learning a new task interferes only minimally with previously learned tasks, and the amount of already used network capacity can be monitored via the singular value spectra and quota of conceptors.

In Jaeger (2014), different scenarios for continual learning are investigated in a reservoir computing setting. Two extreme cases are obtained when (i) the involved learning tasks are entirely unrelated to each other, versus (ii) all tasks come from the same parametric family of learning tasks. The two cases differ conspicuously with regards to the geometry of involved conceptors, and with regards to opportunities to re-use previously acquired functionality in subsequent learning episodes. The permuted MNIST task is an example of (i) while the disjoint MNIST task rather is of type (ii). Conceptors provide an analytical tool to discuss the "family relatedness" and enabling/disabling conditions for continual learning in geometrical terms. Ongoing and future research is devoted to a comprehensive mathematical analysis of these phenomena which in our view lie at the heart of understanding continual learning.

### ACKNOWLEDGMENTS

The work reported in this article was partly funded through the European H2020 collaborative project NeuRAM3 (grant Nr 687299).

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
