# OpenReview forum: "Overcoming Catastrophic Interference using Conceptor-Aided Backpropagation"
_ICLR.cc/2018/Conference — Accept (Poster)_

### Official Review · AnonReviewer1 · 2017-11-26
**This is an interesting method for continual learning. It relies mostly on conceptors, Linear Algebra method, for minimizing the interference of new task to the already learned tasks.**

**Rating:** 7
**Confidence:** 3

**Review:**

This paper introduces a method for learning new tasks, without interfering previous tasks, using conceptors. This method originates from linear algebra, where a the network tries to algebraically infer the main subspace where previous tasks were learned, and make the network learn the new task in a new sub-space which is "unused" until the present task in hand.

The paper starts with describing the method and giving some context for the method and previous methods that deal with the same problem. In Section 2 the authors review conceptors. This method is algebraic method closely related to spanning sub spaces and SVD. The main advantage of using conceptors is their trait of boolean logics: i.e., their ability to be added and multiplied naturally. In section 3 the authors elaborate on reviewed ocnceptors method and show how to adapt this algorithm to SGD with back-propagation. The authors provide a version with batch SGD as well.

In Section 4, the authors show their method on permuted MNIST. They compare the method to EWC with the same architecture. They show that their method more efficiently suffers on permuted MNIST from less degradation. Also, they compared the method to EWC and IMM on disjoint MNIST and again got the best performance.

In general, unlike what the authors suggest, I do not believe this method is how biological agents perform their tasks in real life. Nevertheless, the authors show that their method indeed reduce the interference generated by a new task on the old learned tasks.

I think that this work might interest the community since such methods might be part of the tools that practitioners have in order to cope with learning new tasks without destroying the previous ones.  What is missing is the following: I think that without any additional effort, a network can learn a new task in parallel to other task, or some other techniques may be used which are not bound to any algebraic methods. Therefore, my only concern is that in this comparison the work bounded to very specific group of methods, and the question of what is the best method for continual learning remained open.

---

> ### Author Response · Authors · 2017-12-13
> **Response to Reviewer1**
>
> Thank you for your feedback! We absolutely share your disbelief that „this method is how biological agents perform their tasks in real life“. But we made no such claims - and after re-reading our paper we could not find a spot that could be interpreted as if we viewed our model as biologically relevant. (We want to add in parentheses that we are engaged in a collaboration with a neuroscience group, aiming at revealing dendritic spike dynamics as a possible carrier for biological conceptors; but in our paper we made no allusion to this line of work).
>
> As for your final concern, as we understand it, you point out that biological neural networks are able to cope with a number of different learning tasks simultaneously or in a dovetailing fashion (but we are not sure whether we understand you correctly), and you deplore that we are only comparing to the „very specific group of methods" and problem definitions that are currently considered in the machine learning (ML) community. Yes, in ML only a rather narrow version of continual learning is addressed which one could dub „strictly sequential learning“: first learn task A, then learn B,  etc. Obviously animals and humans can do better and learn (very!) many tasks interleavingly. But strictly sequential learning is difficult enough in ML/ANN research and the catastrophic forgetting problem that it raises hasn’t been satisfactorily addressed until recently. Your suggestion points out a natural and relevant extension of ML research directions!

---

### Official Review · AnonReviewer2 · 2017-11-27
**A stimulating article which in spite of some problems makes people think**

**Rating:** 7
**Confidence:** 3

**Review:**

[Reviewed on January 12th]

This article applies the notion of “conceptors” -- a form of regulariser introduced by the same author a few years ago, exhibiting appealing boolean logic pseudo-operations -- to prevent forgetting in continual learning,more precisely in the training of neural networks on sequential tasks. It proposes itself as an improvement over the main recent development of the field, namely Elastic Weight Consolidation.  After a brief and clear introduction to conceptors and their application to ridge regression, the authors explain how to inject conceptors into Stochastic Gradient Descent and finally, the real innovation of the paper, into Backpropagation. Follows a section of experiments on variants of MNIST commonly used for continual learning.

Continual learning in neural networks is a hot topic, and this article contributes a very interesting idea. The notion of conceptors is appealing in this particular use for its interpretation in terms of regularizer and in terms of Boolean logic.  The numeric examples, although quite toy, provide a clear illustration.

A few things are still missing to back the strong claims of this paper:
* Some considerations of the computational costs: the reliance on the full NxN correlation matrix R makes me fear it might be costly, as it is applied to every layer of the neural networks and hence is the largest number of units in a layer.  This is of course much lighter than if it were the covariance matrix of all the weights, which would be daunting, but still deserves to be addressed, if only with wall time measures.
* It could also be welcome to use a more grounded vocabulary, e.g. on p.2 “Figure 1 shows examples of conceptors computer from three clouds of sample state points coming from a hypothetical 3-neuron recurrent network that was drive with input signals from three difference sources” could be much more simply said as “Figure 1 shows the ellipses corresponding to three sets of R^3 points”. Being less grandiose would make the value of this article nicely on its own.
* Some examples beyond the contrived MNIST toy examples would be welcome. For example, the main method this article is compared to (EWC) had a very strong section on Reinforcement learning examples in the Atari framework, not only as an illustration but also as a motivation. I realise not everyone has the computational or engineering resources to try extensively on multiple benchmarks from classification to reinforcement learning. Nevertheless, without going to that extreme, it might be worth adding an extra demo on something bigger than MNIST. The authors transparently explain in their answer that they do not (yet!) belong to the deep learning community and hope finding some collaborations to pursue this further. If I may make a suggestion, I think their work would get much stronger impact by  doing it the reverse way: first finding the collaboration, then adding this extra empirical results, which then leads to a bigger impact publication.

The later point would normally make me attribute a score of "6: Marginally above acceptance threshold" by current DL community standards, but because there is such a pressing need for methods to tackle this problem, and because this article can generate thinking along new lines about this, I give it a 7 : Good paper, accept.

---

> ### Author Response · Authors · 2017-12-13
> **Response to Reviewer2**
>
> Thank you for your feedback! As to your main concern, i.e. that we dodged the blind submission policies by a previous ArXiv publication, we wish to emphasize that in no way did we want to violate these rules. We were relying on the statement found on the submission webpage (http://www.iclr.cc/doku.php?id=iclr2018:conference_cfp): "While ICLR is double blind, we will not forbid authors from posting their paper on arXiv or any other public forum". If we misunderstood this statement, we apologize and will of course retract our submission; but before we do so, we would want to get a word of guidance from the conference organizers how that statement should be properly interpreted.
>
> We are very grateful that even while you felt, well, cheated by the previous ArXiv publication, you composed an insightful and constructive review. Regarding the computational cost, since a conceptor can be computed by ridge regression, the time complexity is O(nN^2+N^3) if the design matrix is dense, where n is the number of samples and N the number of features. In terms of wall time measures, the time taken to compute a conceptor from the entire MNIST training set (n=55000 images and N=784 pixels, corresponding to the input layer in our networks) is 0.42 seconds of standard notebook CPU time on average. Incremental online adaptation by gradient descent of conceptors is possible in principle too and would come at a cost of O(N^2) per update; we did not implement this. A detailed analysis of computational cost will be added in the revision.
>
> As for your second suggestion (a larger-sized demo), we have to admit that due to lack of resources (time, infrastructure and manpower) we are currently unable to evaluate our method on tasks of the caliber that you suggest. If our method will be well received in the deep learning community (to which we do not really belong), we hope to find cooperation partners in the future to explore larger-than-MNIST tasks.
>
> Finally, we will go through the paper again to make the vocabulary and phrasing more grounded.

---

> > ### Comment · AnonReviewer2 · 2018-01-12
> > **Response from Reviewer 2**
> >
> > Dear Authors
> >
> > Thank you for pointing to the statement on the submission webpage. I agree with your interpretation, and retract my objection: even though I find this utterly confusing (and would have wished that the PC and AC detail this in their request to reviewers), this is not for you to pay the price.
> >
> > Apologies for the stress this may have caused you. I will revise my review.
> > Best regards.

---

### Official Review · AnonReviewer3 · 2017-11-27
**contribution unclear**

**Rating:** 7
**Confidence:** 5

**Review:**

The paper leaves me guessing which part is a new contribution, and which one is already possible with conceptors as described in the Jaeger 2014 report. Figure (1) in the paper is identical to the one in the (short version of) the Jaeger report but is missing an explicit reference. Figure 2 is almost identical, again a reference to the original would be better.
Conceptors can be trained with a number of approaches (as described both in the 2014 Jaeger tech report and in the JMLR paper), including ridge regression. What I am missing here is a clear indication what is an original contribution of the paper, and what is already possible using the original approach. The fact that additional conceptors can be trained does not appear new for the approach described here. If the presented approach was an improvement over the original conceptors, the evaluation should compare the new and the original version.

The evaluation also leaves me a little confused in an additional dimension: the paper title and abstract suggested that the contribution is about overcoming catastrophic forgetting. The evaluation shows that the approach performs better classifying MNIST digits than another approach. This is nice but doesn't really tell me much about overcoming catastrophic forgetting.

---

> ### Author Response · Authors · 2017-12-14
> **Reviewer3 misunderstood the paper**
>
> It seems that the reviewer did not read our paper carefully, since it is clear that this paper is not about improving conceptors per se, but about applying conceptors to overcoming catastrophic interference in neural networks.  The permuted and disjoint MNIST classification tasks used to evaluate our approach are commonly chosen in continual learning literature to demonstrate a method can overcome catastrophic forgetting (for details, see Lee et al., 2017; Kirkpatrick et al., 2017; Kemker et al., 2017 in the References). The basic idea behind these tests is to show that a neural network can still classify the first datasets without catastrophic forgetting after it is trained on other different tasks. Reviewer 3 (and only this reviewer) entirely misunderstood the objectives and contributions of our work.

---

### Decision · Program_Chairs · 2018-01-29
**ICLR 2018 Conference Acceptance Decision**

**Decision:**

Accept (Poster)

**Comment:**

This paper is a timely application of linear algebra to propose a method for reducing catastrophic interference by training a new task in a subspace of the parameter space using conceptors. The conceptors are deployed in the backprop, making this a valuable alternative to recent continual learning methods such as EWC. The paper is clearly written and the results give a clear validation of the method. The reviewers agree as to the merits of the paper.